# Electrical impedance tomography in pulmonary arterial hypertension

**André L. D. Hovnanian**\*, **Eduardo L. V. Costa**, **Susana Hoette, Caio J. C. S. Fernandes, Carlos V. P. Jardim, Bruno A. Dias, Luciana T. K. Morinaga, Marcelo B. P. Amato, Rogério Souza**

Pulmonary Divison, Heart Institute, Hospital das Clínicas, University of São Paulo, São Paulo, Brazil

\* ahovnanian@yahoo.com.br

## Abstract

The characterization of pulmonary arterial hypertension (PAH) relies mainly on right heart catheterization (RHC). Electrical impedance tomography (EIT) provides a non-invasive estimation of lung perfusion that could complement the hemodynamic information from RHC. To assess the association between impedance variation of lung perfusion ($\Delta Z_Q$) and hemodynamic profile, severity, and prognosis, suspected of PAH or worsening PAH patients were submitted simultaneously to RHC and EIT. Measurements of $\Delta Z_Q$ were obtained. Based on the results of the RHC, 35 patients composed the PAH group, and eight patients, the normopressoric (NP) group. PAH patients showed a significantly reduced $\Delta Z_Q$ compared to the NP group. There was a significant correlation between $\Delta Z_Q$ and hemodynamic parameters, particularly with stroke volume (SV) (r = 0.76; P < 0.001). At 60 months, 15 patients died (43%) and 1 received lung transplantation; at baseline they had worse hemodynamics, and reduced $\Delta Z_Q$ when compared to survivors. Patients with low $\Delta Z_Q$ ($\leq 154.6\%$. Kg) presented significantly worse survival (P = 0.033). $\Delta Z_Q$ is associated with hemodynamic status of PAH patients, with disease severity and survival, demonstrating EIT as a promising tool for monitoring patients with pulmonary vascular disease.

## Introduction

Pulmonary arterial hypertension (PAH) is a progressive disease of the pulmonary circulation encompassing an intense vascular remodeling process, leading to severe disruption of vascular mechanics, right ventricle dysfunction and, ultimately, premature death [1, 2]. Right heart catheterization (RHC) remains the most appropriate method for PAH diagnosis with significant prognostic information [2]. Although different imaging modalities provide significant noninvasive information about pulmonary vascular physiology, PAH severity as well as prognosis [3–6], methods for estimation of lung perfusion remain scarce.

Electrical impedance tomography (EIT) is a non-invasive imaging tool that identifies both lung ventilation and perfusion simultaneously based on measurements of thoracic impedance changes [7]. While the entry of air in the lungs causes impedance to increase, because of its low resistivity, the flow of blood into the pulmonary circulation during systole leads to a

**Funding:** This study was supported by FAPESP (Fundação de Amparo à Pesquisa do Estado de São Paulo). http://www.fapesp.br/ The funders had no role in study design, data collection and analysis, decision to publish, or preparation of the manuscript.

**Competing interests:** The authors have declared that no competing interests exist.

decrease in the thoracic impedance signal. Since these two phenomena occur at different frequencies, it is possible to separate the signal of perfusion from that of ventilation [8].

Despite the potential for clinical application, there is limited information about the use of EIT on PAH patients. In one study of eight patients with idiopathic PAH (IPAH), a single patient responded to the vasodilatation test; in this patient, there was correlation between impedance change related to lung perfusion ($\Delta Z_Q$) and the change on pulmonary vascular resistance (PVR) and mean pulmonary artery pressure (mPAP). The authors suggested that EIT reliably measured pulmonary intra-vascular blood volume changes [9]. In another study [10], there was a significantly reduced $\Delta Z_Q$ in IPAH compared to healthy volunteers, probably indicating an impairment of pulmonary vascular mechanics. Our hypothesis is that EIT carries pathophysiological information, reflecting PAH severity.

The main objective of this study was to assess EIT as a noninvasive prognostic imaging modality in PAH through its ability to reflect PAH severity according to RHC findings. Thus, the association between $\Delta Z_Q$ and the hemodynamic profile, disease severity, and survival of PAH patients was evaluated.

## Materials and methods

The study received the approval of the Research Ethics Committee of the Heart Institute, Hospital das Clínicas da Faculdade de Medicina da Universidade de São Paulo, approval number: 1392/06. The form of consent obtained was written.

### Study population and design

We prospectively studied adult patients with suspected PAH or diagnosed PAH referred for invasive hemodynamic evaluation. For those patients under diagnostic evaluation, pulmonary hypertension was suspected by the combination of suggested symptoms/signs plus the finding of systolic PAP $\geq$40 mmHg in the transthoracic echocardiography. All other patients included in the study already had the diagnosis of pulmonary hypertension (Table 1).

Pulmonary arterial hypertension was defined by a resting mPAP $\geq$25 mmHg during RHC, with a pulmonary wedge pressure (PWP) $\leq$15 mmHg, in the absence of significant lung parenchyma disease, left heart dysfunction and chronic thromboembolic disease [2, 11]. The EIT acquisition was performed simultaneously to RHC. Based on RHC findings, patients were discriminated into 2 groups: PAH group and those with normal hemodynamics, named here as "normopressoric" (NP) group.

The study was conducted at the Hemodynamic Laboratory of a tertiary pulmonary hypertension reference center. After the diagnosis of PAH, all the patients were followed periodically on an outpatient basis at the same institution. There were no losses to follow-up.

The protocol was approved by the local ethics committee and all participants gave written informed consent.

### Right heart catheterization

A complete hemodynamic invasive evaluation was performed in all patients using standard techniques for RHC [7]. Hemodynamic measurements included mean right atrial pressure, mPAP, PWP, and cardiac output (CO) determined by the thermodilution technique. Cardiac index (CI) was calculated as CO divided by body surface area. Pulmonary vascular resistance (PVR) was calculated as (mPAP–PWP) divided by CO. Pulmonary vascular compliance (PVC) was calculated as systolic volume (SV) divided by pulse pressure (systolic PAP–diastolic PAP).

**Table 1. Baseline characteristics of the study population.**

| | NP | PAH | p |
|---|---|---|---|
| | (n = 8) | (n = 35) | |
| *Demographics* | | | |
| Sex, Female:Male | 6 (3): 2 (1) | 26 (2.9): 9 (1) | |
| Age, years | 40.1 ± 15 | 42.8 ± 14.5 | 0.37 |
| Weight, Kg.m$^{-2}$ | 62.4 ± 15 | 65.9 ± 16.2 | 0.29 |
| Height, m | 1.56 ± 0.06 | 1.60 ± 0.1 | 0.11 |
| *Functional Class* | | | |
| CF I/II | 8 (100%) | 24 (68,6%) | 0.90 |
| CF III/IV | - | 11 (31,4%) | |
| *Biomarkers* | | | |
| BNP (ng/dL) | 65.6 ± 105.4 | 247 ± 304.3 | 0.006 |
| *Hemodynamics* | | | |
| mPAP, mm Hg | 19.1 ± 4 | 55.5 ± 16.2 | <0.001 |
| PWP, mm Hg | 9.1 ± 3.4 | 10.4 ± 3.1 | 0.15 |
| SV, mL | 75.6 ± 22.6 | 53.7 ± 18.7 | 0.013 |
| CO, L.min$^{-1}$ | 6.4 ± 2.2 | 4.1 ± 1.1 | <0.001 |
| PVR, Woods | 2.3 ± 1.6 | 11.7 ± 6.4 | <0.001 |
| Compl, mL.mm Hg-$^{1}$ | 4.6 ± 2.2 | 1.3 ± 0.9 | <0.001 |
| *Etiologies* | | | |
| IPAH | - | 15 (42.9%) | |
| CTD | 3 (37.5%) | 9 (25.7%) | |
| Schistosomiasis | - | 4 (11.4%) | |
| Portopulmonary | 3 (37.5%) | 3 (8.6%) | |
| HIV | - | 2 (5.7%) | |
| Congenital cardiac shunts | - | 2 (5.7%) | |
| Sickle cell disease | 1 (12.5%) | - | |
| Other | 1 (12.5%) | - | |
| *Treatment* | | | |
| Sildenafil | - | 9 (25.7%) | |
| Bosentan | - | 3 (8.6%) | |
| Combined therapy | - | 5 (14.3%) | |
| Naïve | - | 18 (51.4%) | |

*Definitions of abbreviations*: NP = normopressoric; PAH = pulmonary arterial hypertension; BNP = brain natriuretic peptide; mPAP = mean pulmonary arterial pressure; PWP = pulmonary wedge pressure; SV = stroke volume; CO = cardiac output; NYAH = New York Heart Association; PVR = pulmonary vascular resistance; PVC = pulmonary vascular compliance; IPAH = idiopathic pulmonary arterial hypertension; CTD = collagenous tissue diseases; HIV = human immunodeficiency virus; PDE5i = phosphodiesterase type 5 inhibitors; ERA = endothelin-1 receptor antagonists.

The continuous variables are presented as mean ± standard deviation (SD), when normally distributed, and otherwise as median and interquartile [25–75%] ranges.

## EIT protocol

The EIT measurements and image acquisition were performed with the platform *Enlight 1800*, a 32-electrode device (Timpel, Sao Paulo, Brazil).

**Image acquisition protocol.** The 32-electrodes were attached circumferentially to the surface of the patients' thorax at about the level of the fourth intercostal space. The EKG-gated technique was used in order to eliminate the ventilatory impedance oscillations [12]. Patients remained under resting condition, in supine position. One hundred cardiac cycles were automatically averaged to obtain one complete data set. Image files were recorded for 3 minutes for

posterior off-line analysis. Since the obtained impedance images are relative images, the measurement of $\Delta Z_Q$ are expressed as percent values (%).

**Off-line image analysis.** The recorded file was analyzed using a dedicated software developed in Labview 7.1 (National Instruments, USA). This software was validated in another study from our group [13]. A temporal series of 50 images per second was acquired, each image comprising a 32x32 matrix, in which each value represented a pixel.

To quantify the impedance change within the lungs from the images, we analyzed the images using regions of interest (ROI). ROI analysis was performed using individual masks for each file for both ventilation and perfusion [14]. We built perfusion masks following a stepwise approach, as follows: 1. Cardiac pixels–which include the potential anatomical areas corresponding to both the heart and the great vessels–tend to be out of phase with lung pixels, because the lungs receive the blood ejected during systole. Based on the phase lag of each pixel in the EKG-gated image and a typical lung pixel, cardiac pixels were automatically excluded; 2. Typical lung pixels were determined by the identification of a decrease in $\Delta Z_Q$ value in the $\Delta Z_Q$ vs. time curves immediately after the point corresponding to EKG-gating (Fig 2); 3. The lung mask comprised of all pixels after the exclusion of cardiac pixels; 4. The sum of the mask-derived pixel amplitudes was multiplied by body weight [14] to yield the $\Delta Z_Q$ measure.

## Statistical analysis

The continuous variables are presented as mean ± standard deviation (SD), when normally distributed, otherwise as median and interquartile [25–75%] ranges. The Student t-test and the Mann-Whitney test were used for group comparison, as appropriate. The Fisher exact test was used to compare categorical variables. For the estimation of the correlation between the $\Delta Z_Q$ and hemodynamic parameters, we used the Pearson correlation analysis. $\Delta Z_Q$ was dichotomized in low vs high $\Delta Z_Q$ based on its median value. Survival curves were described using the Kaplan-Meier Product Time Limit method, and the low vs high $\Delta Z_Q$ groups were compared with the Log-rank test. Hazard ratios and 95% confidence intervals were estimated with Cox proportional hazards analysis in a multivariable model in which both $\Delta Z_Q$ and BNP were included. P values <0.05 were considered significant. The analysis was performed using SPSS software, version 17.0) and R (version 3.0.2).

## Results

A total of 55 patients were submitted to RHC. The hemodynamic measurements were normal in eight patients, who represented the normopressoric group. Forty-seven patients received the diagnosis of pulmonary hypertension, of whom 12 were excluded (6 due to EIT signal/file problems; 4 due to PWP >15 mm Hg; and 2 patients due to lack of reliable measure of CO). Thirty-five patients composed the PAH group.

Baseline characteristics of both groups are presented in Table 1. All NP patients were symptomatic at the time of the RHC. Only 30% of the PAH patients had NYHA III/IV, despite a severe hemodynamic profile. The brain natriuretic peptide (BNP) levels were normal in NP group and were consistently elevated in the PAH group. Six etiologies of PAH were identified, of which 70% were composed by idiopathic PAH (IPAH) and connective tissue disease associated PAH (CTD_PAH). More than half of the PAH group was composed of newly diagnosed, treatment-naïve patients.

$\Delta Z_Q$ was significantly lower in the PAH group compared to the NP group (177.8±90.3 vs. 277.8±160.5%.Kg; p = 0.046) (Fig 1). Fig 2 illustrates the typical images of $\Delta Z_Q$ and its respective variation over time for both groups.

$\Delta Z_Q$ correlated with mPAP, PVC, PVR, CO, and especially with SV (Fig 3).

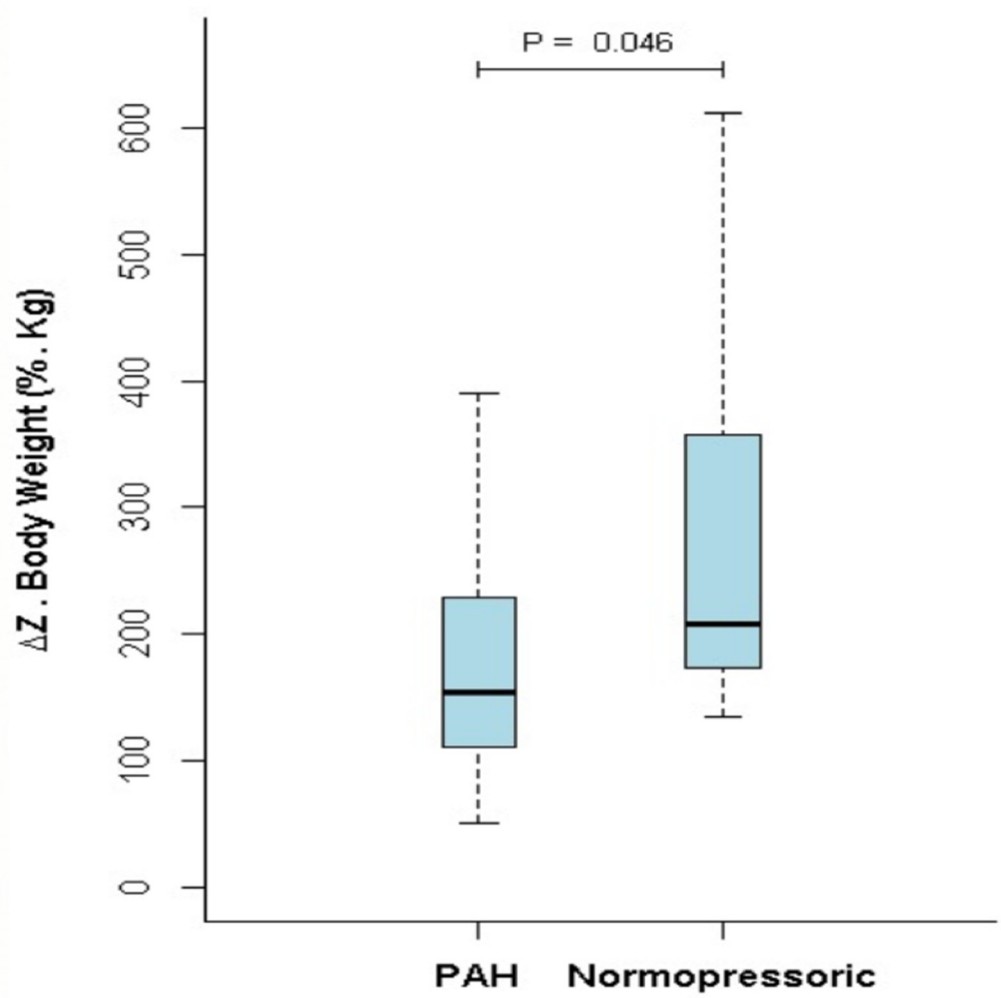

**Fig 1. Comparison between the $\Delta Z_Q$ value of the pulmonary arterial hypertension group and the normopressoric group.**

During follow-up, 15 patients died and one received bilateral lung transplantation successfully. Fourteen of them were female (87%). There was no difference in mean age between survivors and non-survivors. Non-survivors had a worse hemodynamic profile: CO (3.6±1.0 vs. 4.7±1.2 L/min; p = 0.004), SV (44.2±12.1 vs. 63.9±19 mL; p < 0.001), mPAP (60.4±13.7 vs. 47.8±16.8 mmHg; p = 0.01), PVR (14.6±6.6 vs. 8.6±4.6 Woods Unit (WU); p = 0.02), PVC (0.9 ±0.3 vs. 1.8±1.2 mL.mmHg$^{-1}$; p = 0.002); there was no difference in PWP (10.3±3.1 vs. 10.4 ±3.2 mL.mmHg$^{-1}$; p = NS).

$\Delta Z_Q$ was significantly reduced in the non-survivors in comparison to the survivors (127.2 ±54.3 vs. 231.6±91%.Kg; p < 0.001) (Fig 4).

Patients with low $\Delta Z_Q$ (≤154.6%.Kg) also had a worse hemodynamic profile when compared to the patients with high $\Delta Z_Q$: CO (3.8±1.3 vs. 4.5±1.1 L/min; p = 0.037), SV (45.3±15.4 vs. 62.7±18 mL; p = 0.002), mPAP (62.3±12 vs. 45.8±16.3 mmHg; p < 0.001), PVR (15.1±6.5 vs. 8±3.9 WU; p < 0.001), PVC (0.9±0.4 vs. 1.7±1.2 mL.mmHg$^{-1}$; p = 0.004); there was no difference in PWP (9.9±3.8 vs. 11±2.2 mL.mmHg$^{-1}$; p = NS).

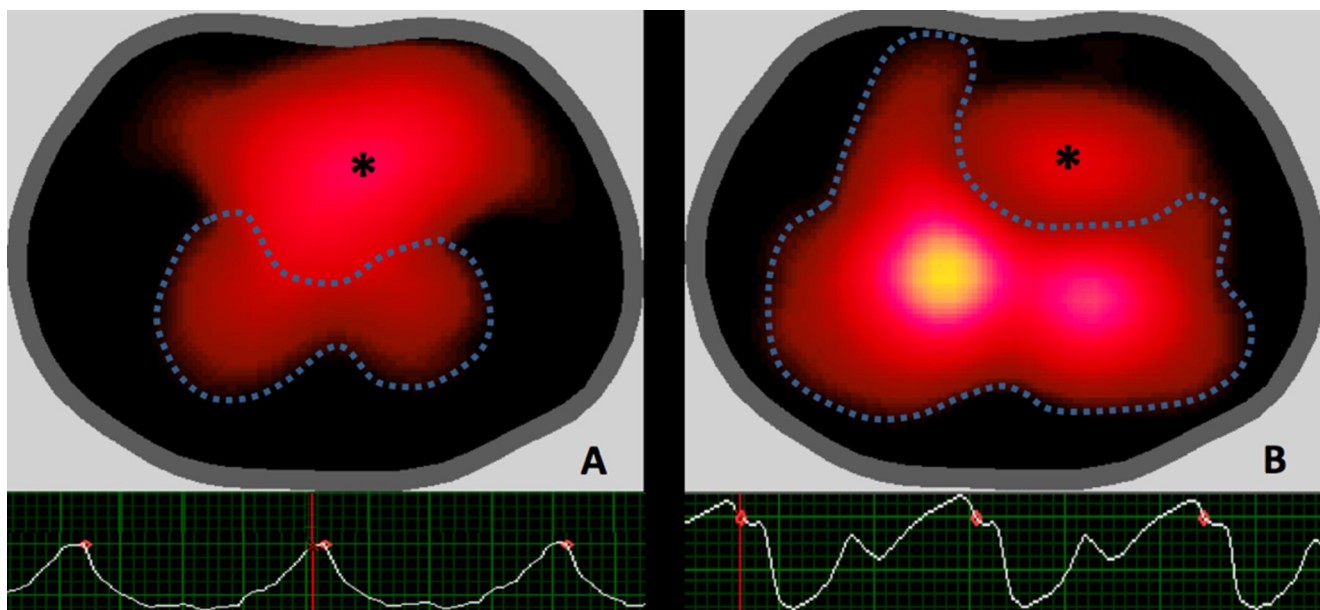

**Fig 2.** EIT images and $\Delta Z_Q$ vs. time curves of the pulmonary arterial hypertension group (A) and of the normopressoric group (B). The dashed lines represent the pixels corresponding to the pulmonary area; the central yellow area corresponds to the pixels with higher $\Delta Z_Q$ values, while the dark-red regions to the pixels with lower $\Delta Z_Q$ values; * refers to the cardiac pixels. Below, the $\Delta Z_Q$ vs. time curves: the red points indicate gating to the QRS complex of the ECG–it correlates with ventricular systole and filling of the pulmonary circulation, which is responsible for the descent of the $\Delta Z_Q$ value; note the difference in the amplitude and waveforms.

BNP measurements were available in 28 patients. In a multivariable analysis, $\Delta Z_Q$ independently predicted survival (HR 0.98, 95%CI 0.97–0.99; p = 0.011) whereas BNP did not (HR 1.0, 95%CI 0.99–1.0; p = 0.928).

The median survival in the low-$\Delta Z_Q$ group was 3.94 years, while median survival was not reached in the high-$\Delta Z_Q$ group. Patients with low $\Delta Z_Q$ ($\leq$154.6%.Kg) had presented a worse survival when compared to the patients with high $\Delta Z_Q$ (p = 0.033) (Fig 5).

## Discussion

The present study analyzed the application of EIT, an emergent imaging modality, to patients with suspected or confirmed PAH. Our results demonstrated that $\Delta Z_Q$ shows a significant association with the hemodynamic profile, the disease severity, and the prognosis of PAH patients.

Seventy-five percent of our PAH group were composed by women, with mean age in the fifth decade of life, and nutritional status in the range of overweight. These findings are in line with the data from recent registries [15–17]. Furthermore, our patients, the majority of whom had IPAH or CTD_PAH, had severe and advanced forms of PAH evidenced by their poor hemodynamic profile.

Patients with PAH presented with a significantly reduced $\Delta Z_Q$ in comparison to the NP individuals. A similar result was presented by a Dutch study, in 2006, which identified an important reduction in $\Delta Z_Q$ in 21 patients with IPAH compared to 30 healthy volunteers [10]. Nevertheless, three different aspects between the two studies are relevant: first, we used a newer generation of EIT technology which comprises a 32x32 pixel matrix, a noncircular mesh and a better image temporal resolution; second, our control group was composed of patients with conditions related to PAH, all symptomatic, who had clinical indication for RHC, but did not confirm the presence of pulmonary hypertension. These patients where

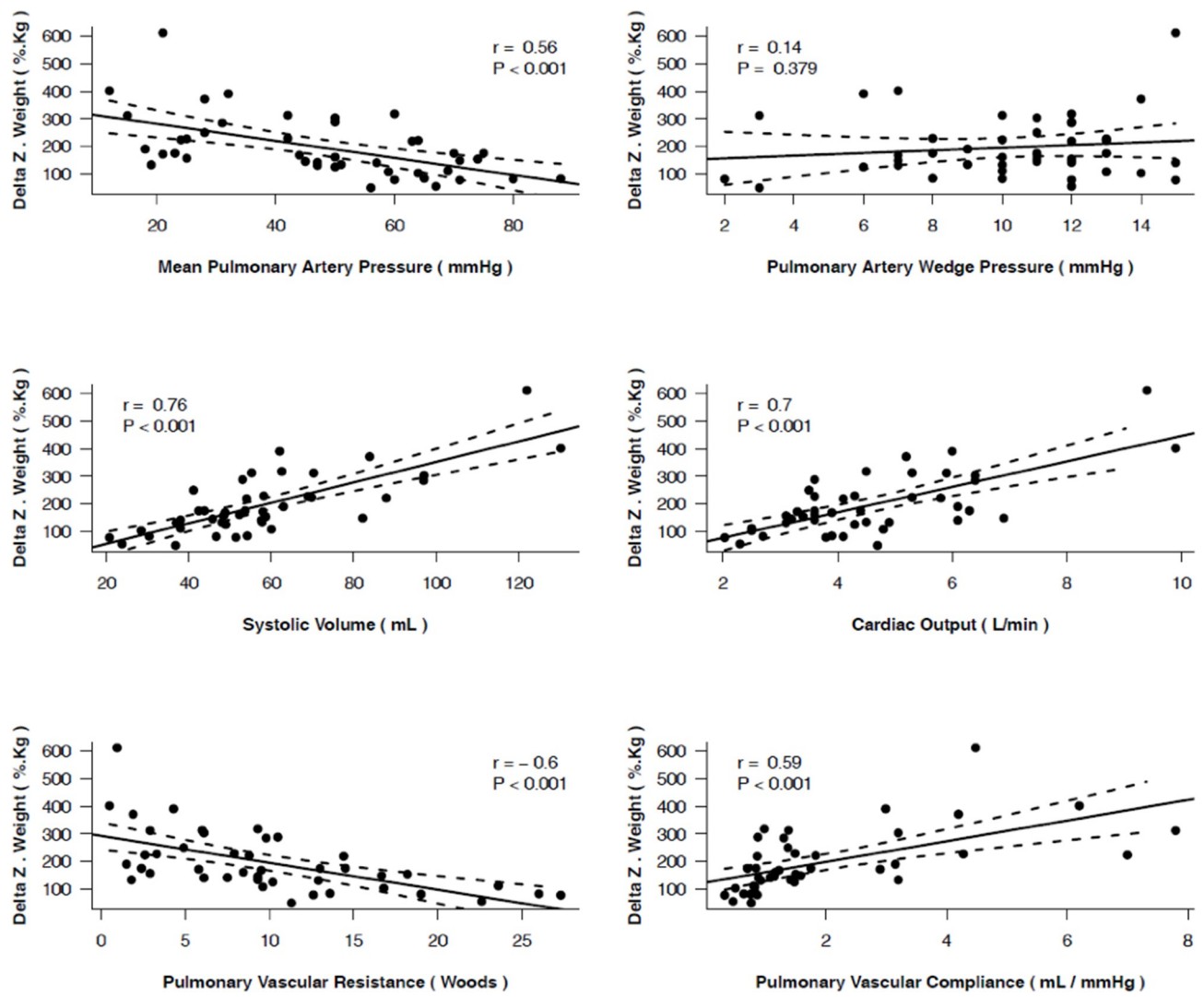

**Fig 3. Correlations between $\Delta Z_Q$ and invasive hemodynamic parameters.**

named here as normopressoric (NP) and represented more than 14% (8/55) of the patients referred to RHC, and more than 30% (8/26) of those who performed the RHC in search for PH diagnosis. In a study from our group, about 20% of the individuals that underwent RHC as part of the diagnostic evaluation for the presence of pulmonary hypertension presented normal pressure levels [18]. Distinctly of health volunteers, it is not possible to assert that the NP patients do not have incipient disease. Even so, there was a significant difference between the PAH and the NP groups [19]. Including patients with indication for RHC rather than healthy volunteers in the control group was important to avoid spectrum bias. Finally, our PAH group was composed of six different forms of PAH whereas only IPAH patients were included in the Dutch study. Our patient mix reflects more closely patients with suspected PAH in clinical practice.

The second relevant finding of our study refers to the correlation of the $\Delta Z_Q$ to the hemodynamic profile. The $\Delta Z_Q$ correlated both to PVC and PVR, mechanical constituents of pulmonary circulation, as well as to SV, the surrogate of right ventricular function. Only one clinical study observed correlation of $\Delta Z_Q$ to hemodynamic parameters in a single patient who

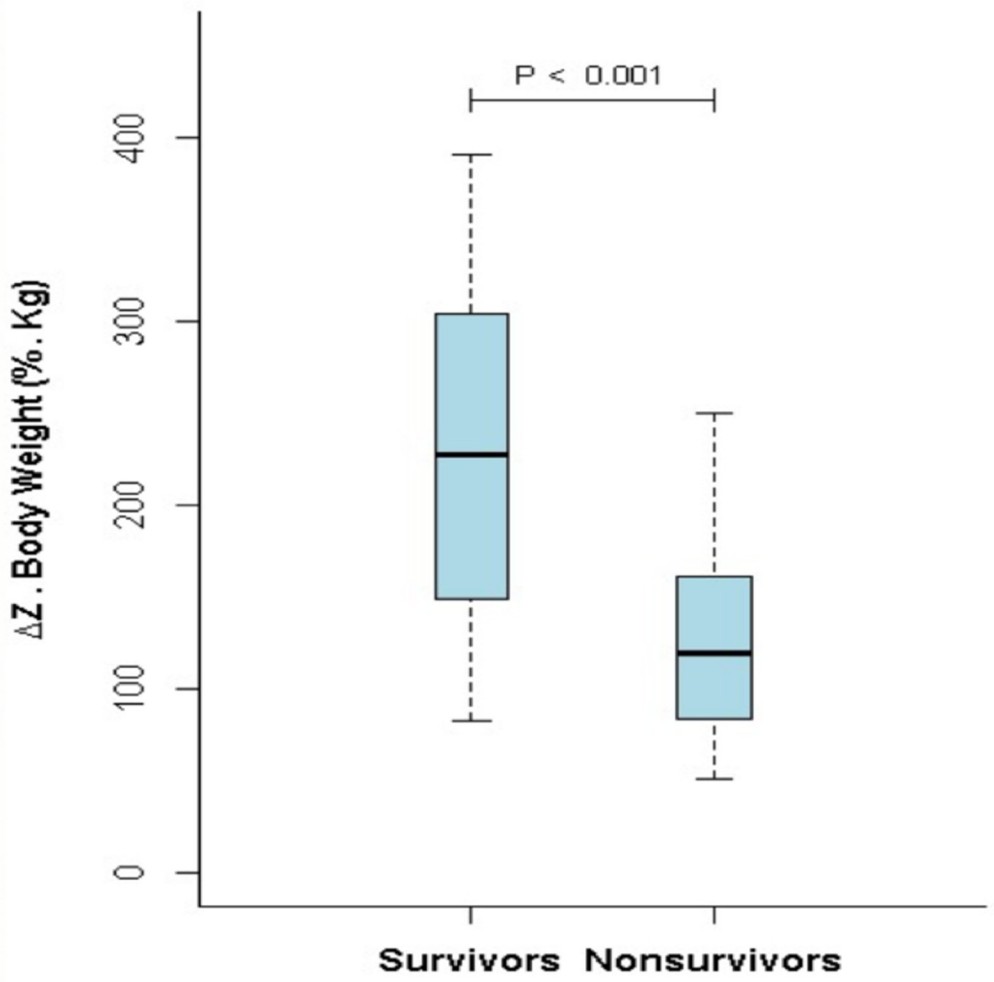

**Fig 4. Comparison between the $\Delta Z_Q$ value of the survivors and the nonsurvivors groups.**

presented a progressive increase in $\Delta Z_Q$ parallel to a decrease in mPAP and PVR during the vasoreactivity test [9]. Correlation between $\Delta Z_Q$ and SV was not evaluated in that study. On the other hand, our study demonstrated significant correlations between $\Delta Z_Q$ and all hemodynamic parameters except for PWP. Such a difference can be explained by two factors: (i) our population was composed of 43 patients with hemodynamic data–to the best of our knowledge, this is the largest clinical investigation of EIT in PAH in the presence of simultaneous hemodynamic data; and (ii) our systematic approach to exclude non-pulmonary pixels [20]. Thus, our measurement of $\Delta Z_Q$ probably reflected more accurately the actual magnitude of the pulmonary intra-vascular blood volume variation, illustrating the so sought non-invasive and functional information about the microvascular pulmonary bed. This finding can place EIT as a promising form of assessing the pulmonary circulation physiology and PAH pathophysiology.

Another noteworthy result of our study is the correlation between $\Delta Z_Q$ and SV. An MRI study demonstrated that indexed SV $<25$ mL.$(m^2)^{-1}$ and indexed right ventricular diastolic volume $>84$ mL.$(m^2)^{-1}$ were significantly associated with worse prognosis in IPAH [5]. Stroke volume has a pivotal role on the severity and survival of PAH patients, especially because it represents the impairment of the right ventricle (RV) due to the pathological changes in

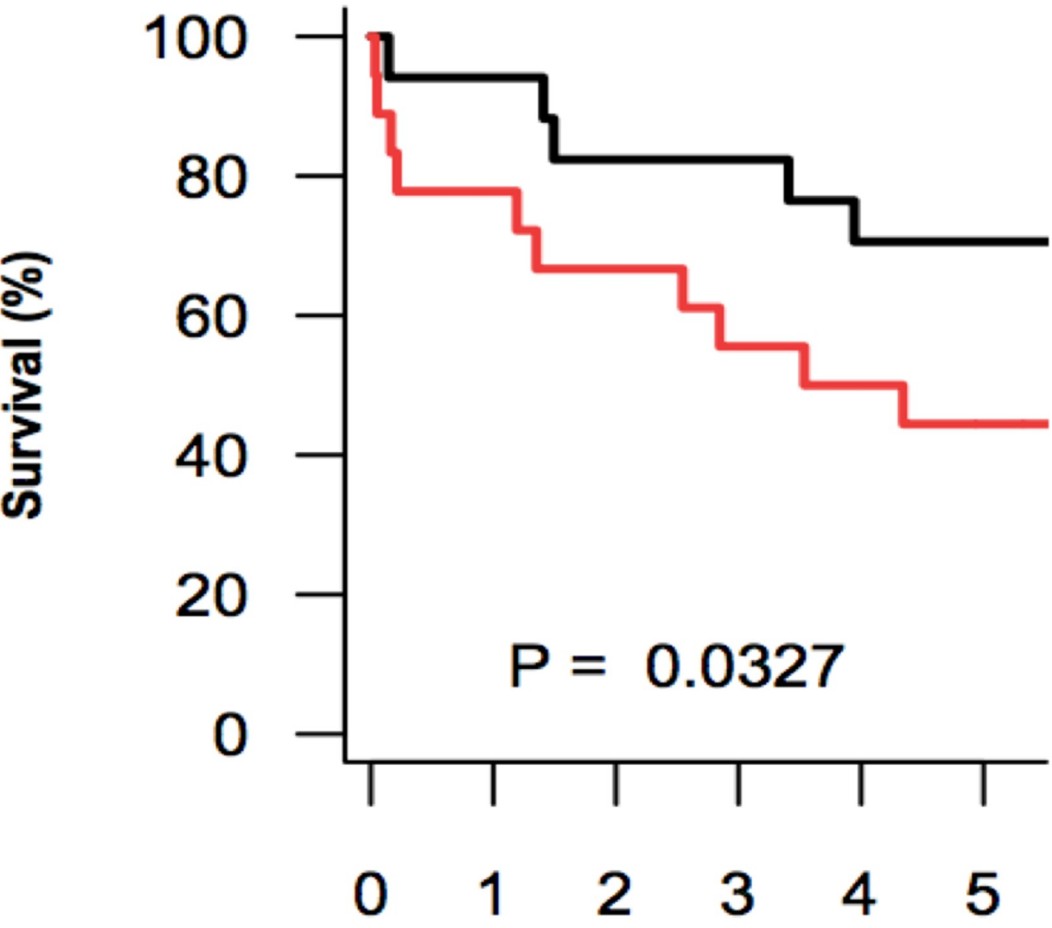

**Fig 5. Kaplan-Meier transplant-free survival estimates in the high-$\Delta Z_Q$ group (black) and in the low-$\Delta Z_Q$ group (red) according to the median of the $\Delta Z_Q$.**

pulmonary vascular mechanics. A recent study [21] suggested that SV provides significant prognostic information even in patients already stratified as presenting low-risk of disease progression. Despite some difficulties reported in estimating SV by means of EIT [22], one can also understand EIT as a non-invasive tool for assessment of disease severity given the good correlation of SV and $\Delta Z_Q$ (r = 0.76; p < 0.001) in our study as well as in an experimental study of our group [14]. This assumption is strengthened by the finding of an even more

pronounced reduction of $\Delta Z_Q$ in the 16 patients who died or received lung transplantation during follow-up, in accordance with a significant worse hemodynamic profile in this group, including a significantly lower SV.

In the current state of PAH management, even after great achievements in the understanding and management, overall mortality in PAH is still elevated [17, 23–25]. In our study, the median survival in the low $\Delta Z_Q$ group was almost four years, and EIT was capable of discriminating those patients with higher risk of death or lung transplantation. Risk assessment is paramount in estimating the prognosis of PAH patients during follow-up [25–27]. Multidimensional scores combining clinical, echocardiographic, biological, hemodynamic, and exercise variables have been used to stratify patients in low or high-risk of death or lung transplantation [25–27]. These scores have used noninvasive parameters in an attempt to improve the prognostic ability of invasive hemodynamic parameters. We showed that electrical impedance tomography is also a non-invasive method that carries prognostic information in PAH patients. Thus, it is reasonable to suggest the incorporation of EIT in future multimodal scores as well as its use for the serial follow-up assessments of PAH patients.

Two aspects related to the EIT assessment are noteworthy. The first was our choice to only include lung pixels on the pulsatility analysis. This approach is better than measuring the heart pulsatility because the heart position in relation to the electrode belt can introduce measurement error [11, 22]. Additionally, lung pulsatility can further decrease in more severe PAH because of lower vascular compliance. The second aspect was the correction to body weight. As well as in a recent experimental study of our group [14], that also observed a strong association between $\Delta Z_Q$ and SV, the correlation between $\Delta Z_Q$ and SV was improved by the incorporation of weight to the $\Delta Z_Q$ value. Since EIT measurements are relative images based on changes in impedance in relation to a reference, the $\Delta Z_Q$ is expressed as a percentage change in impedance in relation to the reference condition. The advantage of this approach is the improvement in the robustness of the image reconstruction. In contrast, it produces outputs that are insensible to the absolute values of impedance. It means that an equal absolute perturbation inside thoraces of different sizes and different muscle-to-fat proportion will produce distinct impedance changes, reinforcing the importance of anthropometric correction.

Additionally, information about reproducibility and feasibility deserves consideration. The reproducibility of EIT has been studied in both ventilation and perfusion studies [22, 28–31]. In a model for estimating SV by means of EIT, compared to cardiac MRI, a mean error of -10 ±12.8 mL was found [30]. Two of these studies [22, 31] showed that the core difficulty in EIT-based SV monitoring is that purely amplitude-based features (that is the case of our study) are strongly influenced by other factors: posture, electrode contact impedance and lung or heart conductivity. The EIT technology consists in installing two auto-adhesive belts over the thorax of a patient by a trained technician. Each belt is connected to a cable that transfers information between the belts and the (trans-)portable EIT device, whose digital/touchable interface also allows a simple operation by a trained technician. These characteristics demonstrate the feasibility of EIT for bedside evaluation in hospitalized patients [32], and probably for the outpatient ones.

Our study has some limitations. First, we did not explore the potential of EIT as a diagnostic tool for pulmonary hypertension, and we only monitored each patient once. Second, there are some limitations related to the EIT technology itself. EKG-gating requires a regular heart rhythm. In patients with arrhythmias such as atrial fibrillation, $\Delta Z_Q$ can be significantly influenced by the ventilation signal. As a result, six patients were excluded because of inadequate image reconstruction or filtering. Additionally, it is known that the pulsatile signal at the cardiac frequency is not purely due to lung perfusion: the impedance changes related to vascular lung pulses can be related to other determinants such as PAP and surrounding lung tissue

compliance [33]. The indicator dilution technique could help in the understanding if our results would express strictly lung perfusion [34].

Finally, there are still many challenges ahead before EIT can be incorporated in the field of pulmonary hypertension. Many gaps of knowledge should be explored in future studies. For example, the usefulness of EIT for the diagnosis of PAH or for monitoring response to PAH specific treatment is yet unknown.

In conclusion, given its close association with the hemodynamic profile, disease severity, and survival of PAH patients, EIT can be seen as a promising non-invasive tool for monitoring patients with pulmonary vascular disease.

## Supporting information

**S1 Data.**
(CSV)

## Author Contributions

**Conceptualization:** André L. D. Hovnanian, Eduardo L. V. Costa, Susana Hoette, Caio J. C. S. Fernandes, Carlos V. P. Jardim, Bruno A. Dias, Luciana T. K. Morinaga, Marcelo B. P. Amato, Rogério Souza.

**Data curation:** André L. D. Hovnanian, Eduardo L. V. Costa, Marcelo B. P. Amato, Rogério Souza.

**Formal analysis:** André L. D. Hovnanian, Eduardo L. V. Costa, Susana Hoette, Caio J. C. S. Fernandes, Carlos V. P. Jardim, Bruno A. Dias, Luciana T. K. Morinaga, Marcelo B. P. Amato, Rogério Souza.

**Funding acquisition:** Rogério Souza.

**Investigation:** André L. D. Hovnanian.

**Methodology:** André L. D. Hovnanian, Eduardo L. V. Costa, Marcelo B. P. Amato, Rogério Souza.

**Project administration:** André L. D. Hovnanian, Rogério Souza.

**Resources:** André L. D. Hovnanian, Marcelo B. P. Amato, Rogério Souza.

**Software:** Eduardo L. V. Costa, Marcelo B. P. Amato.

**Supervision:** Eduardo L. V. Costa, Marcelo B. P. Amato, Rogério Souza.

**Validation:** André L. D. Hovnanian, Eduardo L. V. Costa, Marcelo B. P. Amato, Rogério Souza.

**Visualization:** André L. D. Hovnanian.

**Writing – original draft:** André L. D. Hovnanian, Eduardo L. V. Costa.

**Writing – review & editing:** André L. D. Hovnanian, Eduardo L. V. Costa, Susana Hoette, Caio J. C. S. Fernandes, Carlos V. P. Jardim, Bruno A. Dias, Luciana T. K. Morinaga, Marcelo B. P. Amato, Rogério Souza.

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
