## [Decision Letter · Decision Letter 0]

10 Sep 2020

PONE-D-20-26452

Electrical impedance tomography in pulmonary arterial hypertension

PLOS ONE

Dear Dr. HOVNANIAN,

Thank you for submitting your manuscript to PLOS ONE. After careful consideration, we feel that it has merit but does not fully meet PLOS ONE’s publication criteria as it currently stands. Therefore, we invite you to submit a revised version of the manuscript that addresses the points raised during the review process.

As you can see, one of the reviewers has found a significant number of areas where additional data would be needed to be certain that the outcomes are technically correct - and both reviewers were in agreement that the number of control patients was a significant limitation. The revisions requested may be outside the scope of revision, but you may feel that you can reasonably address them.

We look forward to receiving your revised manuscript.

Kind regards,

James West, PhD

Academic Editor

PLOS ONE

Journal Requirements:

3. Please include your tables as part of your main manuscript and remove the individual files. Please note that supplementary tables (should remain/ be uploaded) as separate "supporting information" files

Reviewers' comments:

Reviewer's Responses to Questions

**Comments to the Author**

1. Is the manuscript technically sound, and do the data support the conclusions?

Reviewer #1: Yes

Reviewer #2: No

2. Has the statistical analysis been performed appropriately and rigorously? 

Reviewer #1: Yes

Reviewer #2: No

3. Have the authors made all data underlying the findings in their manuscript fully available?

Reviewer #1: Yes

Reviewer #2: No

4. Is the manuscript presented in an intelligible fashion and written in standard English?

Reviewer #1: Yes

Reviewer #2: Yes

5. Review Comments to the Author

Reviewer #1: This is an excellent study expanding on a clinical assessment that has limited information available at this time. The study was performed in a scientifically sound manner, and presented in a clear, easy to follow and logical manner. The greatest limitation is the small sample size in the study, but given the novel idea this can be overlooked as the incidnce of the disease is also low.

Reviewer #2: In the present study the authors proposed the use of the electrical impedance tomography for the definition of pulmonary arterial hypertension. Non-invasive identification of PAH is very attractive, since it'd allow to reduce the need of invasive right heart cath. Despite the interesting pathophysiological background, the study suffers of several limitations:

- the limited number of patients, in particular of the "healthy" cohort

- how the patients were selected? were they consecutive? Why patients with normal RHC were referred for the invasive procedure? Were they ill?

- How PH was suspected? Was it based on transthoracic echocardiograhy findings?

- the authors excluded pts with post-cap PH. I'd suggest to compare pre-cap with post-cap since the most intriguing aspect might be the differentiation between vascular abnormalities in pre and post-capillary PH

- Information on feasibility, costs and reproducibility of the test should be provided

- echocardiography represents the screening exam for the identification of PH. I'd appreciate a comparison between the performance of echo and EIT. The superiority/complementarity of the two techniques should be discussed

- EIT much strongly correlates with stroke volume. In PAH cohort stroke volume is obviously reduced compared to healthy subjects. How can the authors exclude that EIT could present similar correlation in other diseases determining reduced stroke volume (i.e. HFrEF, valve diseases, etc)

- since most of patients were treatment naive it'd be interesting to assess the trends of EIT longitudinally after treatment initiation

- for all the limitations above, the authors should consider to completely eliminate the paragraph on outcome. I don't feel it necessary since the goal of the study is mostly focused on diagnosis rather than prognostication

6. PLOS authors have the option to publish the peer review history of their article (what does this mean?). If published, this will include your full peer review and any attached files.

Reviewer #1: **Yes: **Alec S Kellish

Reviewer #2: No

---

## [Author Response · Author response to Decision Letter 0]

3 Dec 2020

We thank the reviewers for all the comments and recommendations on our manuscript.

---

## [Decision Letter · Decision Letter 1]

31 Dec 2020

PONE-D-20-26452R1

Electrical impedance tomography in pulmonary arterial hypertension

PLOS ONE

Dear Dr. HOVNANIAN,

Thank you for submitting your manuscript to PLOS ONE. After careful consideration, we feel that it has merit but does not fully meet PLOS ONE’s publication criteria as it currently stands. Therefore, we invite you to submit a revised version of the manuscript that addresses the points raised during the review process.

The current recommendations are primarily around being more cautious in conclusions, and doing a more thorough job of addressing the remaining barriers to inclusion in clinical process.

We look forward to receiving your revised manuscript.

Kind regards,

James West, PhD

Academic Editor

PLOS ONE

Reviewers' comments:

Reviewer's Responses to Questions

**Comments to the Author**

1. If the authors have adequately addressed your comments raised in a previous round of review and you feel that this manuscript is now acceptable for publication, you may indicate that here to bypass the “Comments to the Author” section, enter your conflict of interest statement in the “Confidential to Editor” section, and submit your "Accept" recommendation.

Reviewer #3: (No Response)

2. Is the manuscript technically sound, and do the data support the conclusions?

Reviewer #3: Partly

3. Has the statistical analysis been performed appropriately and rigorously? 

Reviewer #3: Yes

4. Have the authors made all data underlying the findings in their manuscript fully available?

Reviewer #3: Yes

5. Is the manuscript presented in an intelligible fashion and written in standard English?

Reviewer #3: Yes

6. Review Comments to the Author

Reviewer #3: Dear authors, below are a series of comments regarding the format and content of your manuscript.

1. Be careful with abbreviations, the use of PAH and PH confuses the reader and it must be ensured that the same abbreviation format is used throughout the text (page 3 line 96).

2. A paragraph related to patient follow-up would be appreciated in the methodology section.

3. The importance and need of using EIT is not clear, it is a new tool that requires trained personnel and offline analysis with special software, it is expensive (the disposable belts from Timpel) and in the text you mention that "SV provides significant prognostic information " why is this not enough? I suggest to emphasize the importance of multidimensional evaluation in PAH, to justify the use of this new tool.

4. Considering the limitations mentioned in the manuscript, I personally find your conclusion quite ambitious. Much remains to be known and validated before saying "could be incorporated in the routine evaluation of PAH patients".

Perhaps you could incorporate a paragraph mentioning future challenges and next steps related to the use of EIT for the diagnosis and follow-up of PAH.

They are minor suggestions, congratulations on your excellent work.

7. PLOS authors have the option to publish the peer review history of their article (what does this mean?). If published, this will include your full peer review and any attached files.

Reviewer #3: No

---

## [Author Response · Author response to Decision Letter 1]

20 Feb 2021

We appreciate the reviewer #3 for you contribution to our work.

---

## [Editor Report · Decision Letter 2]

23 Feb 2021

Electrical impedance tomography in pulmonary arterial hypertension

PONE-D-20-26452R2

Dear Dr. HOVNANIAN,

We’re pleased to inform you that your manuscript has been judged scientifically suitable for publication and will be formally accepted for publication once it meets all outstanding technical requirements.

Kind regards,

James West, PhD

Academic Editor

PLOS ONE
---

## [Editor Report · Acceptance letter]

1 Mar 2021

PONE-D-20-26452R2 

Electrical impedance tomography in pulmonary arterial hypertension 

Dear Dr. Hovnanian:

I'm pleased to inform you that your manuscript has been deemed suitable for publication in PLOS ONE. Congratulations! Your manuscript is now with our production department. 

Kind regards, 

on behalf of

Dr. James West 

Academic Editor

PLOS ONE